# Metrology for low cost $CO_2$ sensors applications: the case of Steady-State-Through-Flow (SS-TF) chamber for $CO_2$ fluxes observations

Roger Curcoll[1,2], Josep-Anton Morguí[3], Armand Kamnang[3], Lídia Cañas[4], Arturo Vargas[1], Claudia Grossi[1,5]

[1]Institut de Tècniques Energètiques (INTE), Universitat Politècnica de Catalunya, Barcelona, Spain
[2]Departament d'Enginyeria Química, Universitat Politècnica de Catalunya, Terrassa, Spain
[3]Evolutionnary Biology, Ecology and Environmental Science Department, Faculty of Biology, Universitat de Barcelona, Barcelona, Spain
[4]AIRLAB, Climate and Health Program (CLIMA), ISGlobal, Barcelona, Spain
[5]Departament de Física, Universitat Politècnica de Catalunya, Barcelona, Spain

*Correspondence to*: Roger Curcoll (roger.curcoll@upc.edu)

**Abstract.** Soil $CO_2$ emissions are one of the largest contributions to the global carbon cycle, and a full understanding of processes generating them and how climate change may modify them is needed and still uncertain. Thus, a dense spatial and temporal network of $CO_2$ flux measurements from soil could help reduce uncertainty in the global carbon budgets.

In the present study the design, assembling and calibration of low cost Air Enquirer kits, including $CO_2$ and environmental parameters sensors, is presented. Different type of calibrations for the $CO_2$ sensors and their associated errors are calculated. In addition, for the first time this type of sensors have been applied to design, develop and test a new Steady-State-Through-Flow (SS-TF) chamber for simultaneous measurements of $CO_2$ fluxes in soil and $CO_2$ concentrations in air. Sensor's responses were corrected for temperature, relative humidity and pressure conditions in order to reduce the uncertainty of measured $CO_2$ values and of the following calculated $CO_2$ fluxes based on SS-TF. $CO_2$ soil fluxes measured by the proposed SS-TF and by a standard closed Non-Steady-State-Non-Through-Flow (NSS-NTF) chamber were shortly compared to ensure the reliability of the results.

The use of a multi-parametric fitting reduced the total uncertainty of $CO_2$ concentration measurements by 62% compared with the uncertainty if a simple $CO_2$ calibration was applied, and by a 90% when compared to the uncertainty declared by the manufacturer. The new SS-TF system allows continuous measurement of $CO_2$ fluxes and $CO_2$ ambient air with low cost (~1.2 k€), low energy demand (<5W) and low maintenance (twice per year due to sensor calibration requirements).

## 1 Introduction

Global soils store at least twice as much carbon as Earth's atmosphere (Oertel et al., 2016; Scharlemann et al., 2014), and act as sources and/or sinks for greenhouse gases (GHGs) such as carbon dioxide ($CO_2$), methane ($CH_4$), and nitrous oxide ($N_2O$). The total global emission of $CO_2$ from soils is recognized as one of the largest contributions in the global carbon cycle and is, among others, temperature dependent (Bond-Lamberty and Thomson, 2010a). However, soil respiration is probably the least

well constrained component of the terrestrial carbon cycle (Bond-Lamberty and Thomson, 2010b; Schlesinger and Andrews, 2000) and the degree to which climate change will stimulate soil-to-atmosphere $CO_2$ flux remains highly uncertain (Pritchard, 2011). Continuous measurements of soil fluxes are therefore essential to understand changes in soil respiration of ecosystems in relation to climate variables such as atmospheric temperature. A high temporal and spatial resolution monitoring of $CO_2$ fluxes at sensitive areas could offer useful data both for better understanding the processes at the sources and sinks and thus improving biogenic models (Agustí-Panareda et al., 2016; Randerson et al., 2009). In addition, a complete uncertainty budget of $CO_2$ flux measurements will be essential for the evaluation and correction of global flux models and their associated uncertainties.

Gas interchange between the soil and the lower atmosphere is generally measured as the quantity of gas exhaled from the soil per unit of surface and time ($\mu mol \cdot m^{-2} \cdot s^{-1}$). It can be measured with different techniques, being the most common the Steady-State Through-Flow (SS-TF), also known as open dynamic chamber, and the Non-Steady-State Non-Through-Flow (NSS-NTF) or closed chamber (Pumpanen et al., 2004). In both cases, the $CO_2$ fluxes are measured using a chamber installed on the soil surface. NSS-NTF measurements are based on the rate of $CO_2$ concentration increase within the chamber, while in the SS-TF technique the $CO_2$ efflux is continuously calculated as the difference between the $CO_2$ concentration at the inlet and the outlet under determined hypothesis (Livingston and Hutchinson, 1995). In the case of NSS-NTF flux measurements, calibrated data is not strictly necessary as long as the sensor's calibration does not change during the measurement timespan because the flux is proportional to the slope of the $CO_2$ concentration increase within the chamber. SS-TF based results need high accurate calibration sensors because the absolute value of the measured $CO_2$ concentrations into the chamber are used. A literature survey suggests that generally NSS-NTF may underestimate $CO_2$ fluxes by 4–14% probably due to: i) advective fluxes forced by small pressure gradients between the air into the chamber and outside it; ii) setting configurations, such as the installation depth of the chamber into the soil. No significant difference was observed when fluxes were measured using SS-TF chambers where no pressure gradients are created (Pumpanen et al., 2004; Rayment, 2000).

In recent years, Wireless Sensor Networks (WSN) are increasingly used for real time and high spatial resolution monitoring (Oliveira and Rodrigues, 2011). A WSN is composed of spatially distributed autonomous sensors to monitor physical, chemical or environmental conditions, and to cooperatively pass their data through the network to other locations. WSN can be used for local data recording for later analysis or for continuous transmission in real time to a remote laboratory for synchronous analysis.

So far low-cost sensors for $CO_2$ atmospheric measurements have been largely used in industrial environments and for indoor air quality and ventilation rate studies (Fahlen et al., 1992; Mahyuddin and Awbi, 2012; Schell and Int-Hout, 2001). When low cost sensors are applied at high $CO_2$ concentration areas and/or spots where air concentrations observed are in the order of thousands of parts per million (ppm), the total uncertainty of the measurement does not affect the quality of the study of the concentration variability under different conditions and sources/sinks. However, in the last decade, the improvement in precision and cost decrease of Non Dispersive InfraRed (NDIR) $CO_2$ sensors have made them more useful for multiple purposes (Yasuda et al., 2012). Their low weight and dimensions allow their utilization in a wide variety of applications,

including Unmanned Aerial Vehicles (Kunz et al., 2018), $CO_2$ measurements network areas (Kim et al., 2018; Song et al., 2018) and for the study of the distribution of $CO_2$ in large regions, as in the case study of Switzerland (Müller et al., 2020). However, in order to be able to use these sensors in the outdoor atmosphere, a metrological effort is needed to: i) ensure a traceable and stable calibration; ii) evaluate and correct the influence of the environmental parameters, such as temperature, relative humidity and pressure, on the sensor response; iii) estimate the total uncertainty related with the sensors calibrations and corrections.

This work presents a low cost Air Enquirer Kit, including NDIR $CO_2$ and environmental parameters sensors and suggests new possible applications of it to reduce the cost and the maintenance of continuous $CO_2$ fluxes. The manuscript presents the results of the comparison of different calibration methodologies for NDIR $CO_2$ sensors. Furthermore, a new SS-TF system, based on 5 multi-sensors portable Air Enquirer Kits, is presented and shortly compared with a NSS-NTF system at a Spanish mountain site. The system has been designed and built to continuously monitor soil $CO_2$ fluxes with high temporal resolution, high accuracy and low cost and maintenance. This system also allows continuous measurements of ambient $CO_2$ concentration. The SS-TF is made by four Air Enquirer Kits fully characterized under laboratory conditions. The new prototype of the SS-FT chamber is also introduced after describing its theoretical basis as well as the NSS-NTF method. Finally, the results of the sensors calibrations and corrections and of the short NSS-NTF/SS-TF chambers comparison are presented and discussed together with further research steps.

## 2 Methods

### 2.1 Air Enquirer Kit

A multi-sensor portable kit, named Air Enquirer (Morguí et al., 2016), was designed and built in the mark of an EduCaixa project (www.educaixa.org). The kit consists of 5 low cost sensors controlled by an Arduino DUE Rev3 microcontroller board that measure: i) NDIR $CO_2$ concentration (in ppm); ii) relative humidity (%); iii) temperature ($^0C$); iv) barometric pressure (hPa) and v) light intensity (lux). Data from sensors are automatically read and stored at a frequency of 0.2Hz in a microSD card. All sensors and the Arduino board controlling them are enclosed in a methacrylate box of 15x8x5 $cm^3$ in size (Fig. 1). Table 1 shows the main features of each sensor, following specifications provided by their respective manufacturers. The total cost of each Air Enquirer (AE) kit is about 200€.

### 2.2 Calibrations and multi-parametric correction of the $CO_2$ sensors of the Air Enquirer kit

Low-cost $CO_2$ sensors are known to be temperature (T), humidity (H) and pressure (P) dependent (Arzoumanian et al., 2019; Martin et al., 2017). In this study, five AE kits were calibrated using different methodologies from the literature and their responses were corrected under different climate conditions. The simultaneous use of the $CO_2$ and the environmental parameters sensors allows a continuous correction of the response of the $CO_2$ sensor under different conditions of T, P and relative humidity (RH).

First of all, a theoretical correction of the $CO_2$ data was applied taking into account: i) the change from ppm of $CO_2$ in wet air to ppm of $CO_2$ in dry air following Wagner and Pruß, (2002); ii) the conversion from ppm of $CO_2$ measured under specific pressure to the declared using the ideal gas law equation.

The concentration of $CO_2$ in dry air ($CO_{2\_dry}$) was calculated by Eq. (1):

$$CO_{2\_dry} = \frac{CO_{2\_wet}}{V_{dry}} \frac{1013}{P} \tag{1}$$

being $V_{dry}$ the Volume of 1m$^3$ of dry air at 1013 hPa after removing the water volume. $V_{dry}$ can be calculated from Eq. (2):

$$V_{dry} = \frac{P - (P_{ws} \cdot \frac{RH}{100})}{P} \tag{2}$$

being $P_{ws}$ the water vapour saturation directly calculated from Eq (3):

$$P_{ws} = A \cdot 10^{\left(\frac{m \cdot T}{T + T_n}\right)} \tag{3}$$

$A$, $m$ and $T_n$ are constants with values 6.1164, 7.5914 and 240.73 respectively.

In a second step, an experimental multiparametric calibration of the $CO_2$ sensors was done using the data of the environmental sensors and a reference $CO_2$ instrument. A Picarro G2301 Cavity RingDown Spectroscopy Analyzer (CRDS) was used as a second reference standard. This CRDS has a precision better than 0.03 ppm for $CO_2$ (Crosson, 2008; Richardson et al., 2012).

The CRDS results were previously corrected for water vapour (Rella et al., 2013) and calibrated in the laboratory using six NOAA WMO-CO2-X2007 reference gases (primary standard) before and after each experiment following Tans et al. (2011). In order to calibrate the $CO_2$ sensors response for a wide range of temperature, pressure, humidity and $CO_2$ concentration, duplicate measurements were carried out using a temperature controlled box at two sites: i) at the Institut de Ciències del Clima laboratories (IC3), located at 20 meters above sea level (m.a.s.l.), in the city of Barcelona, Spain, and ii) at the Centre de

Recerca d'Alta Muntanya laboratories (CRAM, mountain town of Vielha, Spain, at 1582 m.a.s.l.). Each experiment lasted 7 days and was carried out using the scheme in Fig. 2. In order to remove high frequency variability, the sampled air was homogenised in a sealed pre-chamber prior to entering in the calibration chamber. Then, the air was pumped to the calibration box at a flow rate of 0.4 L·min$^{-1}$ and through the secondary standard reference instrument CRDS. Both experiments were performed in a temperature range between 20 ºC and 42 ºC and a relative humidity with diurnal cycles between 10% and 50%.

Temperature in the calibration box was set to be in increased in slopes of 10ºC, although at low temperatures it fluctuated with room temperature. The pressure ranged between 1004 hPa and 1012 hPa in the calibration at IC3 and between 838 hPa and 850 hPa in the calibration at CRAM. The two calibration experiments at the CRAM and at IC3 stations were carried out with one month difference.

$CO_2$ concentration values measured by each NDIR $CO_2$ sensor and corrected for P and RH using Eq. (1) ($CO_{2\ dry\_kit}$), were

calibrated by comparison with simultaneous $CO_2$ concentration measured by the CRDS ($CO_{2\ CRDS}$) and considering the environmental conditions of T, absolute humidity (H) and P using Eq. (4):

$$CO_{2\ dry\_kit} = \alpha + \beta CO_{2\ CRDS} + \gamma T + \delta H + \varepsilon P \tag{4}$$

A multiparametric fit of Eq. (4), yields the following calibrated/corrected $CO_2$ values as reported in Eq. (5):

$$CO_{2\ corr} = \frac{-\alpha}{\beta} + \frac{1}{\beta} CO_{2\ dry\_kit} - \frac{\gamma}{\beta} T - \frac{\delta}{\beta} H - \frac{\varepsilon}{\beta} P \tag{5}$$

The $CO_{2\ corr}$ calibrated results were compared to those obtained with a simple bias correction using the averages of $CO_{2\ CRDS}$ and $CO_{2\ dry\_kit}$ values and also to those obtained with a simple linear calibration of the $CO_{2\ dry\_kit}$ values with the $CO_{2\ CRDS}$ values without taking in consideration the effect of T, P and H.

**2.3 Steady-State Through-Flow chamber (SS-TF or Open Dynamic Chamber)**

The prototype of the open SS-TF chamber consists of two methacrylate cells of 36 L, where two AE kits are installed in each
of the chambers in order to continuously monitor the $CO_2$ concentration and environmental variables. The duplicity of the AE kits is used to ensure the reliability of the measurements. The chamber dimensions were designed to avoid border effects and minimize measurement errors, as observed by Senevirathna et al. (2007). The first chamber is a hermetic closed chamber with a unique entry for ambient air (labelled here as *Mixing chamber* in Fig. 3). The second one (labelled here as *Flux* chamber), with an open base, was installed directly over the soil.

The *Mixing chamber* is used to mix the sampled air and to measure the $CO_2$ concentration background of the atmospheric air ($C_{mix}$) before it enters into the *Flux chamber*. It contains two AE kits and a fan located at its top for mixing the sampled air. This chamber has only two openings for the inlet and outlet of atmospheric air at a flow of 6.5 L·min$^{-1}$ (labelled $'q'$ in Fig. 3). Cable glands are used at the openings to prevent leakages. Using this configuration, high frequency variability of atmospheric air could be avoided and near steady-state conditions were reached.

The *Flux chamber* is bottomless and has to be positioned in the first 5 cm of the soil/vegetation layer where the soil fluxes are to be measured. Two AE kits and a vent fan were installed at the top of this chamber as well. A constant flow $q$ between the two chambers was achieved with a membrane KNF pump and a flowmeter (labelled as FM in Fig. 3). Low flows, in comparison with the chamber volume, are needed to maintain near steady-state conditions during measurements.

Using the system depicted in Fig. 3, $CO_2$ fluxes ($f_{CO2}$ in µmol·m$^{-2}$·s$^{-1}$) can be calculated for given time intervals within the
*Flux chamber* using the mass balance in Eq. (6) (Gao and Yates, 1998), where, $V$ and $A$ are, respectively the volume of the *Flux chamber* and the emitted soil surface area, $C_a(t)$ (µmol·L$^{-1}$) is the spatially averaged concentration of target gas in the chamber headspace, $C_{in}(t)$ (µmol·L$^{-1}$) is the average $CO_2$ concentration of inlet air in the flux chamber, $C_{out}(t)$ (µmol·L$^{-1}$) is the outflow $CO_2$ concentration, $J_g$ is the flux of the target gas at the enclosed soil surface and $q_{in}$ and $q_{out}$ are the inlet and outlet flow, respectively.

$$dM\ (t) = VdC_a(t) = AJ_g(t)dt + q_{in}C_{in}(t)dt - q_{out}C_{out}(t)dt \tag{6}$$

Assuming that for each measurement interval: i) the inflow and outflow rates are constant and equal (meaning no leakages present in the pneumatic circuit), thus $q_{in}=q_{out}=q$; ii) chamber reach a steady state condition, thus $C_{in}(t)= C_{in}$, $C_{out}(t) = C_{out}$ and $dM(t) = 0$, $CO_2$ flux can be calculated for each time interval from the simplified Eq. (7):

$$f_{CO_2} = J_g = \frac{q}{A}(C_{out} - C_{in}) \tag{7}$$

Assuming that the fan completely mixes the air within the chamber and  the $CO_2$ concentration at each of the boxes is homogeneous, outflow concentration is equal to *Flux chamber* concentration ($C_{out}(t) = C_a(t)$), measured by the two AE kits within the *flux chamber*) and inflow concentration is equal to the mixing concentration ($C_{in}(t) = C_{mix}(t)$), measured by the two AE kits within the *Mixing chamber*. The advantage of this system is that fluxes can be measured continuously with a very small energy requirement (<5 W) and, even using duplicate sensors, with a relative low cost (~1.2k€) in comparison with other

automatic commercial flux chambers, priced at roughly 12 k€. The new system described here enables the feasibility of a network of continuous measurements and a replication of experiments to cope with soil flux variability.

## 2.4 Non-Steady-State Non-Through-Flow chamber (NSS-NTF)

$CO_2$ fluxes using the NSS-NTF chamber or closed static chamber are measured on the basis of the so-called linear accumulation method (Livingston and Hutchinson, 1995), which uses the initial rate of concentration increase in an isolated chamber that

has been placed on the soil surface for a known period of time. Assuming ideal gas behaviour, the slope of the $CO_2$ concentration during the accumulation interval can be used to determine the $CO_2$ flux ($\mu mol \cdot m^{-2} \cdot s^{-1}$) following Eq. (8):

$$f_{CO_2} = J_g = \frac{CO_{2\_slope} \cdot P \cdot V}{A \cdot T \cdot R} \tag{8}$$

where $V$ ($m^3$) and $A$ ($m^2$) are the volume of the chamber and the enclosed soil surface area respectively, $CO_{2\_slope}$ ($ppm \cdot s^{-1}$) is the slope of the linear increment of the $CO_2$ concentration during the early accumulation time, P and T are the atmospheric

pressure and the environmental temperature within the chamber, and R ($m^3 \cdot Pa \cdot K^{-1} \cdot mol^{-1}$) is the universal gas constant. It has been underlined that the linear approach of the accumulation method is only reliable for short time periods (Davidson et al., 2002; Grossi et al., 2012; Gutiérrez-Álvarez et al., 2020). Otherwise, gradients of environmental parameters between the inside and outside chamber could influence the measurement, probably yielding to leakages of unknown origin in the chamber. Luckily, high frequency measurements, as the ones performed by $CO_2$ sensors, allow to apply this method over a really short

accumulation time (T = 5 min has been used in the present study), thus complying with the theoretical requirements. A necessary condition for the application of this method is that the initial $CO_2$ concentration within the chamber has to be equal to the atmospheric $CO_2$ concentration. Therefore, NSS-NTF chambers need to be ventilated after each measurement period (Davidson et al., 2002; Xu et al., 2006). This can be done manually or using automatic systems. In this study, a manual static chamber was used. A closed NSS-NTF chamber of methacrylate (25x25x25) $cm^3$ was built at IC3 in order to perform a short

campaign for the comparison of $CO_2$ fluxes measured by NSS-NTF and SS-TF systems.  An AE (#03) and a fan were fastened at the top of the chamber. Both devices were run by a small external battery pack.  An outer metallic sleeve was previously

fixed onto the soil to avoid leaks and other disturbances. However, the systemic comparison between these two systems is beyond the scope of this study.

## 3 Results and discussion

### 3.1 Comparison between different calibration/corrections approaches

The calibration and correction factors from Eq. (5) of the $CO_2$ sensors installed in the five AE kits are shown in Table 2. The average bias (in ppm $CO_2$) between the AE kit $CO_2$ value after and before applying the theoretical corrections for P and dry air is also shown. The last five columns of Table 2 present, for the different methodologic approaches, the calculated Root Mean Square Error (RMSE) using Eq. (9):

$$\sqrt{\frac{\sum_{i=1}^{n}(x_i^p - x_i^k)^2}{n}} \tag{9}$$

where n is the number of values, $x_i^p$ are the $CO_2$ values of the calibrated CRDS and $x_i^k$ are the $CO_2$ values of the AE sensor for each case: $kCO_2$: uncalibrated values; $kCO_{2\_dry}$: values corrected only for P and dry air; $kCO_{2\_dry-bias}$: values corrected for P and dry air and with the average bias from the CRDS data removed; $kCO_{2\_linear}$: values corrected for P and dry air and linearly calibrated with the CRDS data; $kCO_{2\_multi}$ values corrected for P and dry air and calibrated with the CRDS data using a multiparametric correction with T, RH and P sensors data.

A single theoretical correction for P and RH is demonstrated that already reduces uncertainty by a factor of 5. However, this theoretical correction is not enough for applications where the absolute $CO_2$ value is needed (e.g. for atmospheric composition or SS-TF measurements), as the bias value is extremely variable depending on the sensor unit and up to 50 ppm. When we remove the average bias between the sensor response corrected for P and RH and the CRDS $CO_2$ reference value, the uncertainty is highly reduced and the RMSE of the corrected values ranges between 5.4 ppm and 10.8 ppm. This uncertainty, however, could still be too high for certain applications such as the measurements of small atmospheric variability or for small $CO_2$ fluxes measurements both for the SS-TF and NSS-NTF chambers

Calibrating these sensors through comparison with the CRDS secondary standard in the laboratory by linear fit allows reaching $RMSE_{simple}$ values between 4.2 ppm and 10.9 ppm. However, when the influence of the environmental parameters in the response of the sensors is also taken into account, the $RMSE_{multi}$ values range is shifted to the interval between 2.19 ppm and 5.92 ppm, the lowest ones. Figure 4 shows timeseries of the differences between the $CO_2$ CRDS data and all $CO_2$ sensors data after applying the simple calibration ($CO_{2\_linear}$) and the multiparametric regression ($CO_{2\_multi}$). Corresponding values of T and RH measured during the calibration experiments are also reported. Each $CO_2$ sensor responses differently to the variations of T and RH, and so does the parametric coefficients. Therefore, a theoretical correction of the $CO_2$ value for these variables won't be applicable, and a specific multiparametric fitting is needed.

Figure 5 shows the relation between the reference $CO_2$ values (CRDS) and the values measured by the $CO_2$ sensors both for raw data than after the application of the different calibration methodologies. Four sensors show $RMSE_{multi}$ values lower than

5 ppm, and just one of them (kit #04) greater than 5 ppm. However, this last sensor shows negative correlation with the ambient temperature, unlike all the others where the $CO_2$ values increased as temperature went up. Despite this kit was lately installed within the $CO_2$ fluxes chamber for the second part of the study, results from it were not used.

A variance and covariance analysis were also performed to check the influence of meteorological parameters on the $CO_2$ sensor response. A clear influence of temperature (T), absolute humidity (H) and pressure (P) was observed on the $CO_2$ sensor's response (p-value: $< 10^{-6}$ for all variables). No cross-correlation was observed among variables. It is important to remark that although the multiparametric calibration was done after applying the theoretical correction for P and RH, as explained previously, pressure conditions have the highest influence on the sensor response. In fact, a reduction of 62% in the RMSE was observed when pressure correction was applied. Moreover, parametric values for P diverge between sensors, so every sensor seems to be differently influenced by atmospheric Pressure.

### 3.2 Comparison between the NSS-NTF and SS-TF systems

The new prototype of the SS-TF system, described in section 2.2, was shortly tested in a grassland area of the Pyrenees, near CRAM, between the 1st and the 2nd of June of 2016 and compared with a manual NSS-NTF system. $CO_2$ fluxes ($f_{CO_2}$) were calculated for both SS-TF and NSS-NTF systems, using Eq. (7) and Eq. (8), respectively.

$CO_2$ concentrations from each of the sensors installed in the SS-TF chamber (upper panel) and the corresponding calculated $f_{CO_2}$ time series (lower panel) are shown in Fig. 6. The differences between the ten minutes average of $CO_2$ concentrations measured by the two sensors within the *Mixing chamber* (AE Kits #1 and #2) were of 2.2 ±5.3 ppm. This difference is coherent with the RMSE_multi of both sensors, and remains stable over time. The differences between the ten minutes average of $CO_2$ concentrations measured by the two sensors within the F*lux chamber* (AE Kits #3 and #4) were greater (20 ± 8 ppm) and temperature dependent with a significant correlation (p-value$<10^{-16}$ and $r^2$=0.95). As $CO_2$ values of kit #4 were found to have a different behaviour during the calibration events and the RMSE_multi was greater than 5 ppm, values of this kit were discarded.

Each value of flux has been calculated using Eq. (7) and averaging the calibrated $CO_2$ values of AE #1 and #2 for the mixing chamber and using the calibrated $CO_2$ values from AE #3 for the flux chamber. 10 min. averages were calculated from every minute calculated flux data. The variability of the flux within the 10 minutes averages is represented in Fig. 6 as an associated uncertainty of 2σ. The associated expanded uncertainty for each value has been calculated propagating the 2*RMSE_multi of the flux chamber $CO_2$ sensor.

$CO_2$ fluxes using the NSS-NTF chamber were calculated using the slope of the increase of the $CO_2$ concentration within the chamber and its associated uncertainty. Two examples of the $CO_2$ concentrations measured by the $CO_2$ sensor of kit #03 within the NSS-NTF chamber (see section 2.3) are shown in Figure 7. Data of the first minute after manually closing the chamber were discarded during the $f_{CO_2}$ calculations in order to remove installation noise. Concentration gradients were linear over the following 5 minutes, with a correlation coefficient $R^2$ >0.99 in all cases as calculated with Eq. (5). Positive fluxes were

measured during the afternoon and negative ones at morning as expected because of the photosynthesis phase of grassland plants.

The correlation between both NSS-NTF and SS-TF $f_{CO_2}$ results during the co-measurements carried out at CRAM grasslands during the 1st and the 2nd of June of 2016 is shown in Figure 8. $CO_2$ flux values change from close to zero up to 8 $\mu mol \cdot m^{-2} \cdot s^{-1}$. The obtained $f_{CO_2}$ values agree with $CO_2$ flux values observed in other studies in grasslands at a similar altitude, latitude and period of the year, where the range of night-time fluxes was reported to be between 2 and 4 $\mu mol \cdot m^{-2} \cdot s^{-1}$ (Bahn et al., 2008; Gilmanov et al., 2007). Although the short duration of this first comparison experiment, results help to strengthen the reliability of the new SS-TF chamber based on low cost sensors. However, the size of the comparison dataset does not allow a robust statistic and further long-term comparison should be carried out to fully characterize this new system. Indeed, the main goal of the present manuscript is not characterized the new SS-TF chamber but to offer a robust metrology for low cost $CO_2$ sensors and AE kits which can be easily applied for continuous $CO_2$ flux measurements with high precision, low cost and low maintenance.

CO2 fluxes observations from NSS-NTF and SS-TF chambers agree for positive $CO_2$ fluxes while they do not for negative $CO_2$ fluxes. A plausible cause of this mismatch may be the different degree of opacity of the chambers which influence the sink effect of the soil during the sunlight hours. In fact, the NSS-NTF chamber was completely translucid while in the SS-TF chamber the top side was opaque.

### 3.3 Calibration and recalibration strategy

According to the RMSE results shown in Table 2, the multiparametric correction reduces the uncertainty of $CO_2$ measurements by a factor of 10 compared to those where only a theoretical correction for RH and P was applied and by a factor of 3 compared to a lineal calibration for $CO_2$. In the SS-TF, the flux calculation depends on the difference between the absolute concentrations values of different sensors in two chambers, and a bias between them of e.g. 10 ppm will cause, in this system, a fixed bias of 0.32 $\mu mol \cdot m^{-2} \cdot s^{-1}$ in the flux calculus. Therefore, the multiparametric correction of sensors for this application is strongly recommended, together with a periodical recalibration of the $CO_2$ sensors. Previous works with NDIR sensors have shown that at least every 6-months may be necessary to calibrate the sensors in order to take into account possible effects due to dust and soiling on their internal mirrors (Curcoll et al., 2019; Piedrahita et al., 2014) or the degradation of the IR light (CO2Meters, 2013). A mobile second reference standard could be displayed to perform in situ calibration of the low cost sensors. However, a periodical full calibration and calculation of correction factors for all environmental parameters could be difficult to carry out at field sites, and may even cause large errors if the range of temperature, humidity and pressure used is not large enough. For those cases where a full multiparametric recalibration couldn't be performed each six months, a bias correction should be performed at least every six months. This could be done by placing $CO_2$ sensors in a mixing chamber at the same time and introducing air from a reference tank with known $CO_2$ concentration. Thus, taking in consideration the Eq. 4, this calibration will only adjust the $\alpha$ parameter, considering the effects of P, T and RH constant over the time.

For NSS-NTF applications, where only the slope of the $CO_2$ concentration is used, the bias has no effect on the calculus of the soil flux. Therefore, for this last case periodical corrections for the low cost sensors are not needed although they are advisable to improve the quality of the measurements. Finally, when no calibrations are possible, the recommendation is to calculate the $CO_2$ concentration in dry air and compensate for pressure. Actually, comparing NSS-NTF based flux data, only a difference of about 4% is observed when theoretical correction for P and RH or multiparametric calibration data are compared. However, when using the $CO_2$ AE kits values without any correction this difference rises up to a 23 %.

## 4 Conclusions

Nowadays the improvement in precision and cost decrease of Non Dispersive InfraRed (NDIR) $CO_2$ sensors have made them more readily available for multiple purposes. However, in order to apply them for atmospheric measurements where low $CO_2$ concentrations or small $CO_2$ variability is observed a robust metrology is still needed to: i) ensure a traceable calibration; ii) evaluate and correct the influence of the environmental parameters on the sensor response; iii) estimate the total uncertainty related with the measurements.

In this study an analysis of different calibration methods is carried out for NDIR low cost $CO_2$ sensors using Air Enquirer kits, designed and built, including also environmental sensors. In addition, a new application of these sensors is presented to continuously measure $CO_2$ fluxes on soil with a dynamic chamber.

The lowest uncertainty for the $CO_2$ sensors was obtained by calibrating them using a secondary standard reference (CRDS monitor) and correcting the sensors response under different temperature, humidity and barometric pressure conditions. A multiparametric fitting was applied to calibrate and correct the sensor's responses, achieving a drastic reduction of 90% in the uncertainty of measured $CO_2$ concentrations. The multiparametric calibration will ensure the highest quality of the data and it will be advisable for SS-TF based $CO_2$ flux measurements or $CO_2$ atmospheric concentrations. For NSS-NTF based $CO_2$ flux measurements, a correction for P and RH of the $CO_2$ sensors will already give reliable results, although calibrating the sensors with a portable second reference standard is recommended.

The presented SS-TF chamber based on Air Enquirer kits allows continuous measurement of $CO_2$ fluxes from soil and continuous ambient air $CO_2$ concentration with low uncertainty, low cost (~1.2 k€), low energy demand and low maintenance. This system could be a good tool for creating $CO_2$ flux dense networks. In the present study it has only been shortly compared with a NSS-NTF chamber at Pyrenees area, showing $CO_2$ fluxes comparable between them and in agreement with the literature. However, a full characterization of this system needs to be carried out in the future by long-term comparison with commercial $CO_2$ flux systems.

## Code availability

The software code for this paper is available from the corresponding author.

**Data availability**

The data for this paper are available from the corresponding author.

**Author contributions**

Josep Anton Morguí coordinated the design and manufacture of the AE kits, and promoted the building of the new low cost SS-TF chamber for $CO_2$ fluxes. Lidia Cañas collaborated in the mounting and tuning of the AE kits. Armand Karrang, during his bachelor degree project, participated in the laboratory and field campaigns. Roger Curcoll and Claudia Grossi, performed the laboratory and field experiments, analysed the data and coordinated the manuscript writing. Arturo Vargas participated in the development theoretical approach of the SS-TF methodology for gas fluxes. All authors participated in the data analysis, discussion of the results and writing of the manuscript.

**Competing interests**

The authors declare that they have no conflict of interest.

**Acknowledgements**

The design of the AE Kits, as well as the calibration experiments of the $CO_2$ sensors, were funded by an EduCaixa grant from the CaixaBank Foundation (Principal Investigator (PI): Josep Anton Morguí). The open SS-TF chamber prototype was designed and build at the IC3 in the framework of the project 'Methane interchange over the Iberian Peninsula' and funded by the Retos 2013 grant #CGL2013-46186-R, from the Spanish Ministry of Economy and Competitiveness (PI: Claudia Grossi). The analysis of the data and the preparation of the manuscript was possible thanks to the funding of the Project 19ENV01 traceRadon. This project has received funding from the EMPIR programme co-financed by the Participating States and from the European Union Horizon 2020 research and innovation programme.

Authors would like to thank the Universitat de Barcelona for the use of the CRAM facilities and the team of the Climadat Project (CaixaBank Foundation) at IC3 for support during the laboratory experiments.

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

| Measurement (Units) | Manufacturer | Accuracy | Range of measurement | Operating Temperature (˚C) | Operating Relative Humidity (%) |
|---|---|---|---|---|---|
| $CO_2$ (ppm) | $CO_2$ Engine K30 STA – Sense Air | ±30 ppm$CO_2$ | 0 to 5000 | 0 to 50 | 0 to 95 |
| Temperature (˚C) | DS18B20 – Dallas | ±0.5ºC (within range -20 - +85ºC) | -55 to +125 | -55 to +125 | - |
| Relative Humidity (%) | SparkFun HTU21D – Measurement Specialities | ±2% (within range 20-80%) | 0 to 100 | - 40 to +125 | 0 to 100 |
| Barometric pressure (hPa) | Adafruit *BMP180 - Bosch* | ±1.0 hPa | 300 to 1100 | - 40 to +85 | - |
| Light intensity (visible/IR) | TSL2561 – T.A.O.S. | - | - | - 30 to 70 | 0 to 60 |

**Table 1. Characteristics of the sensors included within the Air Enquirer kit.**


| Kit (code) | Intercept $-\alpha/\beta$ | $CO_{2\_Pic}$ $1/\beta$ | T (ºK) $-\gamma/\beta$ | H (ppm) $-\delta/\beta$ | P (hPa) $-\varepsilon/\beta$ | Bias (ppm $CO_2$) | Root Mean Square Error (ppm $CO_2$) | | | | |
|---|---|---|---|---|---|---|---|---|---|---|---|
| | | | | | | | $kCO_2$ | $kCO_{2\_dry}$ | $kCO_{2\_dry-bias}$ | $kCO_{2\_linear}$ | $kCO_{2\_multi}$ |
| #01 | 59.15 | 1.1047 | -0.395 | $-6.2 \cdot 10^{-4}$ | -0.084 | -9.5 | 76.0 | 12.2 | 7.6 | 7.0 | 3.6 |
| #02 | 52.53 | 1.0564 | -1.594 | $-1.04 \cdot 10^{-3}$ | -0.083 | 51.4 | 43.7 | 52.1 | 8.4 | 8.4 | 2.8 |
| #03 | 93.22 | 1.1031 | -1.150 | $-1.05 \cdot 10^{-3}$ | -0.131 | 21.0 | 57.8 | 23.6 | 10.8 | 10.9 | 2.4 |
| #04 | 49.26 | 1.0908 | 1.306 | $-5.5 \cdot 10^{-4}$ | -0.139 | 1.8 | 68.6 | 9.8 | 9.6 | 10.0 | 5.9 |
| #05 | 13.55 | 1.1030 | -0.570 | $-1.17 \cdot 10^{-3}$ | -0.048 | 14.9 | 58.0 | 15.9 | 5.4 | 4.2 | 2.2 |

**Table 2. Parametric fitting for calibration of $CO_2$ Air Enquirer sensors**



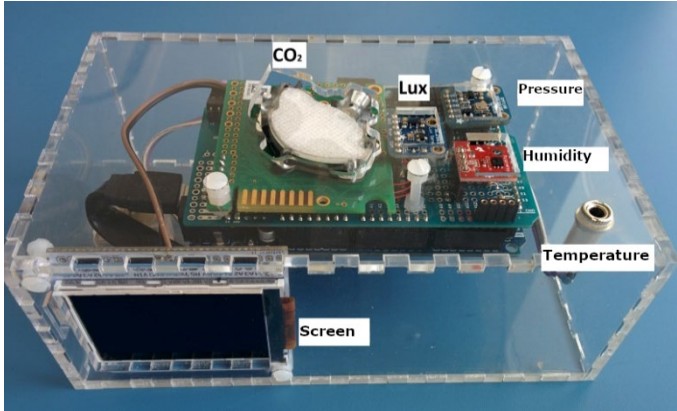

**Figure 1. Air Enquirer kit, with sensors for measurements of temperature, humidity, barometric pressure, light intensity and $CO_2$ concentration in air.**

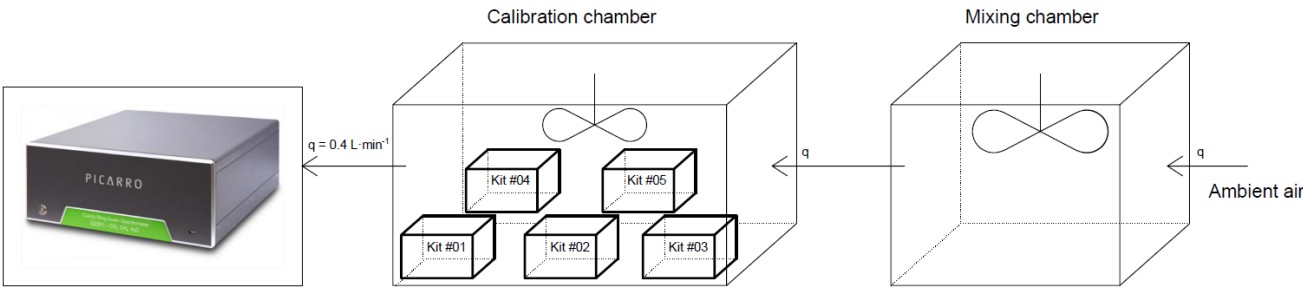

**Figure 2. System used at IC3 (Barcelona, Spain) and at the CRAM station (Vielha, Spain) for the calibration of $CO_2$ sensors mounted on the Air Enquirer kits.**

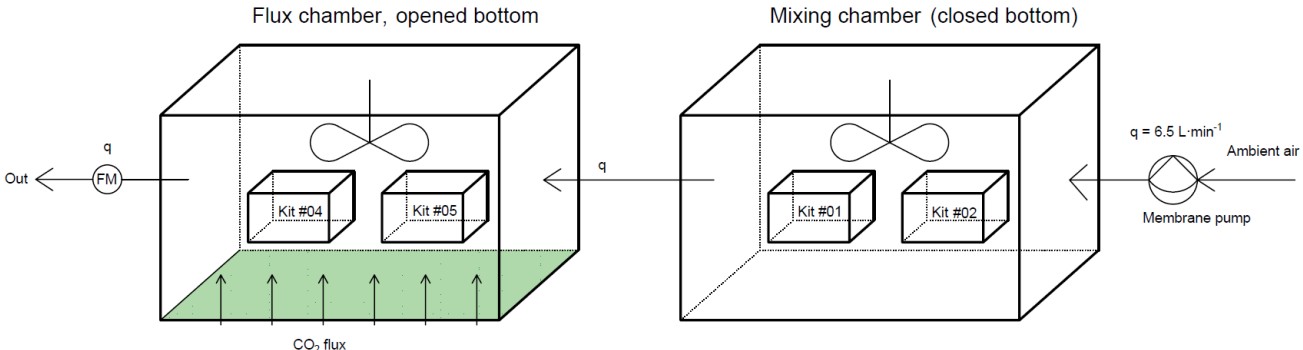


**Figure 3. Scheme of the Dynamic SS-TF Chamber designed and built at IC3 for continuous $CO_2$ flux measurements.**

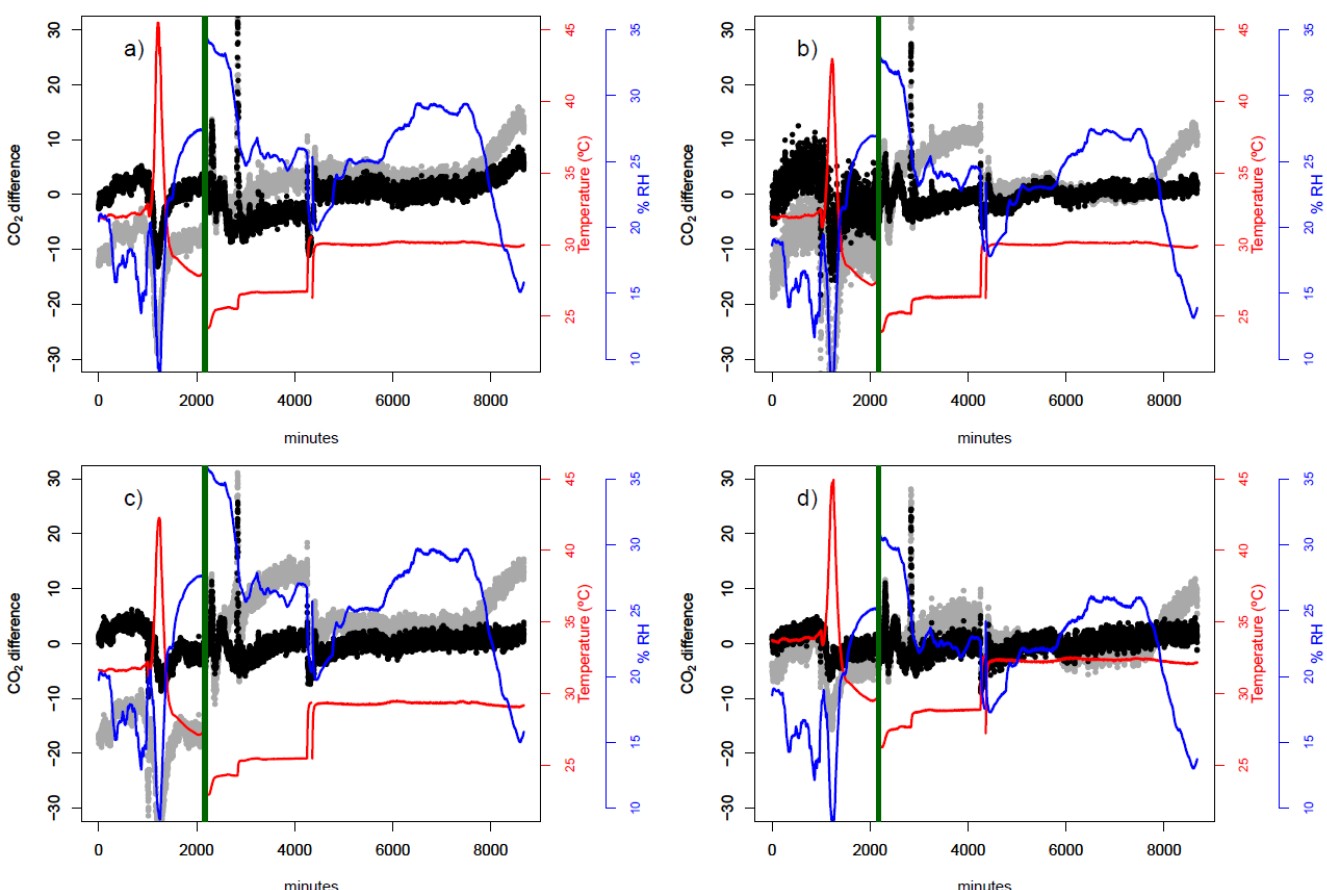

**Figure 4. Timeseries of differences between CRDS CO$_2$ value and CO$_2$ AE kits value after simple calibration (grey) and**
**after multiparametric fitting (black) for AE kit #1 (a), kit #2 (b), kit #3 (c) and kit #5 (d). Temperature values (red) and**
**RH values (blue) are also plotted. Values before vertical green line correspond to the calibration at IC3, and after it to**
**the calibration at CRAM.**


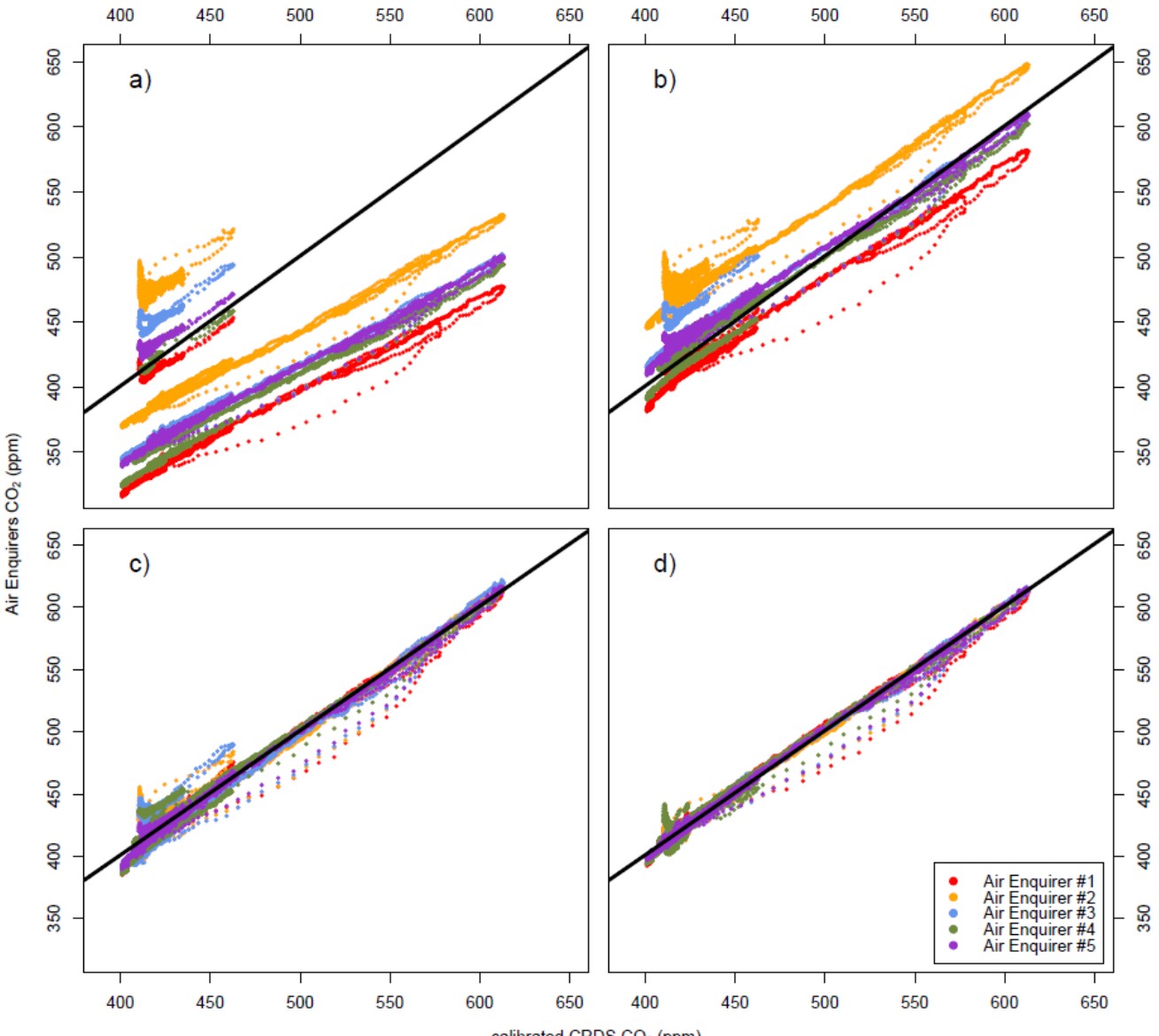

**Figure 5. CO$_2$ concentrations in air measured by each of the AE sensors during the experiment carried out at the CRAM and IC3 stations vs CRDS data using sensor with raw data (a), sensor data theoretically corrected by P and RH (b), sensor data corrected by P and RH and calibrated with the CRDS (c) and sensor data corrected by P and RH and calibrated using a multiparamentric lineal model (d)**

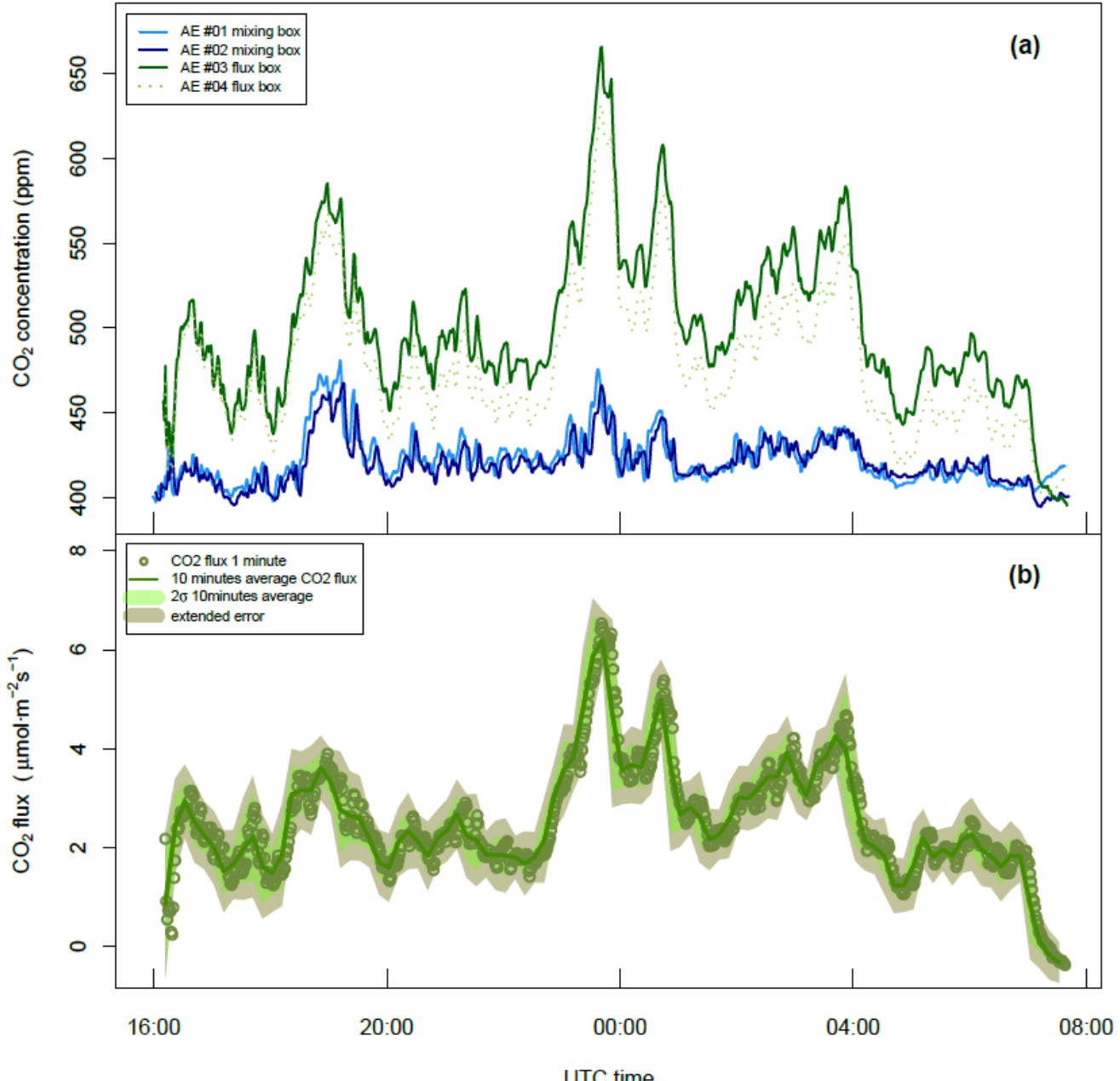

**Figure 6. Time series of 10-min average $CO_2$ concentrations (upper panel) measured within the SS-TF chamber at the CRAM grassland between 1st and 2nd of June 2016, and calculated $f_{CO_2}$ (lower panel). The 2σ range for 10 minutes average variability and the extended error (adding 2 times the RSE of the multiparametric fit) are also plot.**

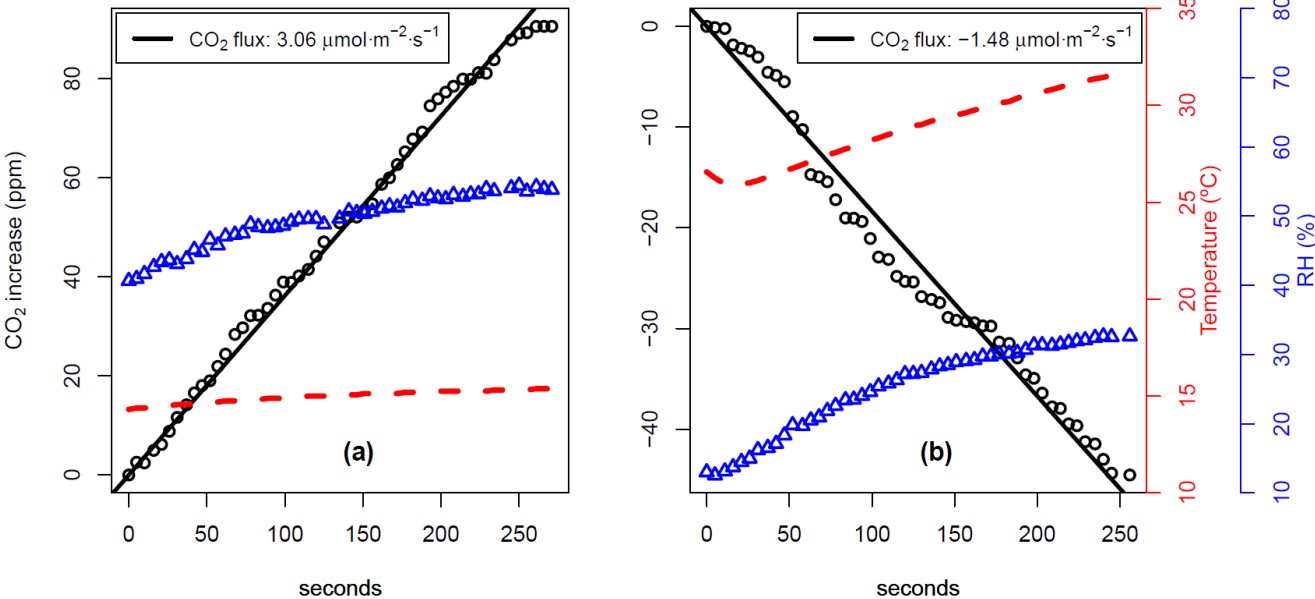

**Figure 7. Example of two cases where the linear accumulation method was applied within an NSS-NTF chamber to calculate positive (a) and negative (b) CO₂ fluxes with Kit #03.**

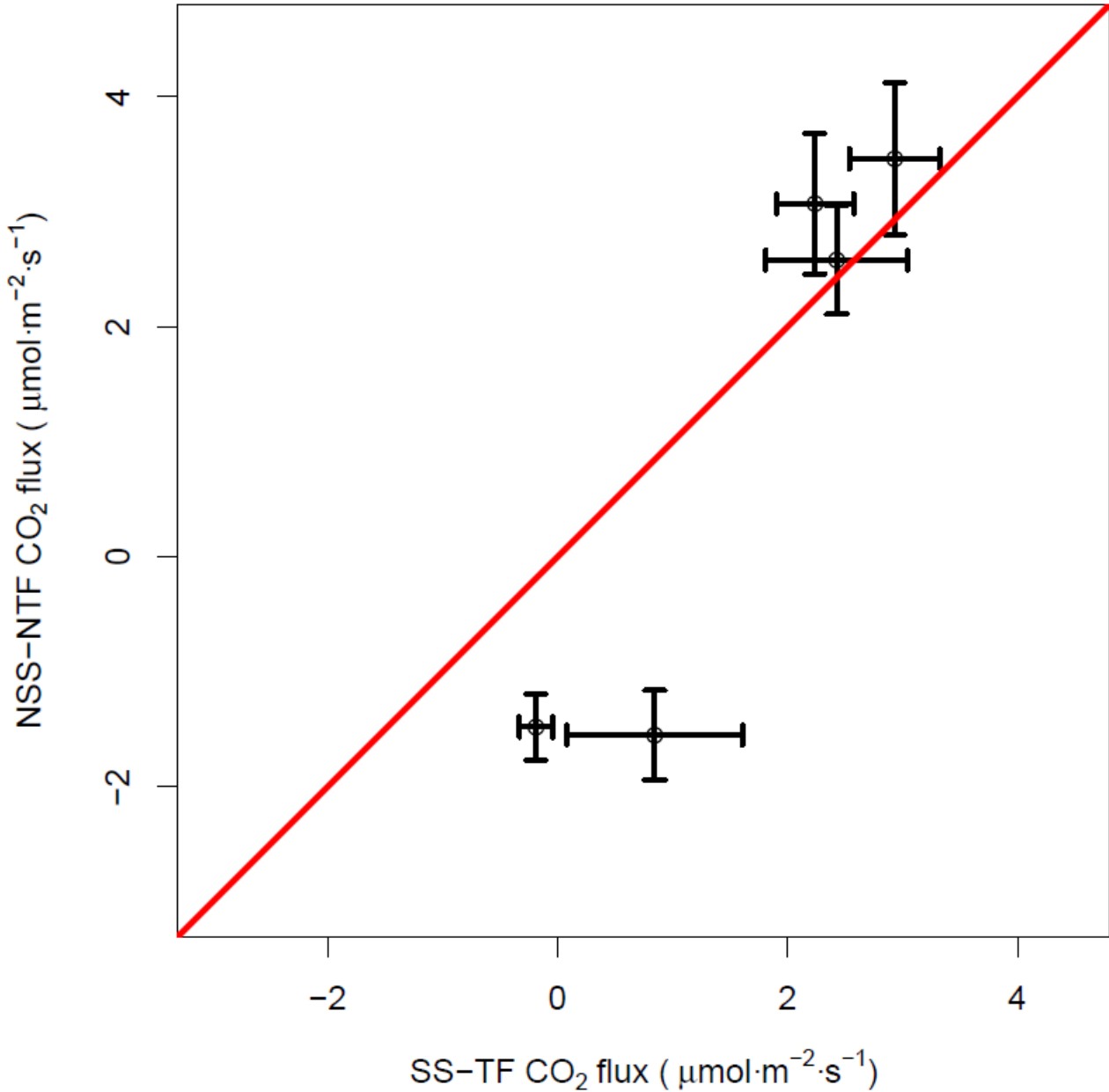

**Figure 8. Comparison of SS-TF and NSS-NTF CO₂ fluxes during a short campaign at the CRAM station between 1st and 2nd of June 2016.**
