# Peer review of "Metrology for low cost CO2 sensors applications: the case of Steady-State-Through-Flow (SS-TF) chamber for CO2 fluxes observations"

_Atmospheric Measurement Techniques, 2021_

## Referee Comment (RC1)

**Reviewer Report on amt-2021-387**

Simone Baffelli, simone.baffelli@empa.ch

**General Comments:**

I appreciate the idea of developing a lower-cost CO2-flux measurement system and find the paper to be well written and of appropriate length and depth

The methods are well documented, as are the results, although I wish some more details on the calibration were shown. On the one side, I find the lack of reference/calibration data for the CO2 flux measurement part a bit disappointing. I wish the authors could provide a comparison with a commercially available flux measurement system, but I understand that organizing such an experiment is not easy. On the other side, I have doubts regarding the CO2 measurement calibration model and the necessity of a relatively complex calibration setup for the particular application. I will discuss these points in more details in the following section.

**Specific Comments**

1) L17: You wrote that the results were corrected for illumination. This does not appear anywhere in the main body of the article. Please either remove this or edit section 2.2 and the results section to reflect this. Do you think illumination would have an impact on the CO2 calibration? I could only imagine an indirect effect through heating of the sensor, which is already captured by the temperature calibration.

2) L88: In my experience employing very similar sensors (SenseAir LP8), air humidity has a large effect on the data quality, particularly for RH> 80, causing an highly nonlinear saturation in the measured concentration. Could you please report the RH range used for calibration? As you measure directly above soil, I would expect the RH in the chamber to routinely reach these values. Did you experience this? This could have an impact on data quality, particularly if you experience a sharp increase in RH within the chamber during the measurement time. Could you show the graph of RH for the linear accumulation experiment of Figure 6?

3) I appreciate the effort spent to develop a high-quality calibration model. However, in the case of NSS-NTF flux measurements, calibrated data is not strictly necessary as long as the sensor's calibration does not change *during* the measurement timespan as the flux is determined "differentially" by considering the rate of growth of the CO2 concentration within the chamber. Could you briefly comment on this in your paper? This could save significant resources for researchers that intend to reproduce your system but do not have the means to perform a comprehensive calibration.

4) In contrast to the above point, in the case of SS-TF chambers, the effect of calibration is not negligible here. For example, if the sensor in the mixing chamber shows a large bias in its response compared to the sensor in the flux chamber (or vice versa), the bias will induce a similar bias in the estimated flux. Therefore, it is clear that uncalibrated sensors cannot be used in this configuration. Could you comment on this? Could you suggest a simplified calibration procedure for those who do not have access to calibration facilities? Would a field calibration with a known gas sample be sufficient?

5) L167 could you provide a summary of the environmental parameters the sensors were exposed to during the calibration cycle? Did you systematically sweep a space of T/RH configurations?

6) L175 could you also show the response of the sensors as a timeseries of relative deviation from the reference provided by the CRDS? Showing this timeseries alongside T and RH would provide some insights regarding their effect on data quality as mentioned in question 3.

7) L183 you report that pressure has the largest effect of the sensor response and that pressure compensation provides the largest reduction in RMSE. Why did you choose to use a linear model for pressure compensation instead of using the quadratic formula suggested by SenseAir in their application note? (see http://www.co2meters.com/Documentation/AppNotes/AN149-Senseair-Pressure-Dependence.pdf)

8) L185 While your statement is true in our experience with the CarboSense network; I think such frequent calibrations could be replaced by in-field calibration for intercept/sensitivity using one respectively two references gases. In the case of NSS-NTF measurements, calibration is not *strictly* necessary to obtain corrected flux measurements as I pointed out above.

9) L202 For the SS-TF measurements you employed multiple sensors on each chamber. Do you think this redundancy could be exploited to improve data quality, for example by averaging the signals or to determine biases?

---

## Author Comment (AC1)

Answer to Reviewer 1:

*General comments:*
*I appreciate the idea of developing a lower-cost CO2-flux measurement system and find the paper to be well written and of appropriate length and depth. The methods are well documented, as are the results, although I wish some more details on the calibration were shown. On the one side, I find the lack of reference/calibration data for the CO2 flux measurement part a bit disappointing. I wish the authors could provide a comparison with a commercially available flux measurement system, but I understand that organizing such an experiment is not easy. On the other side, I have doubts regarding the CO2 measurement calibration model and the necessity of a relatively complex calibration setup for the particular application. I will discuss these points in more details in the following section.*

Dear Simone Baffelli,

First of all, we want to thank you for reviewing this manuscript and for your appreciation of the work done. We found your comments really useful and they have helped us to improve the quality of the manuscript.

Regarding the lack of a comparison with a commercially available flux measurement system, we would like to underline that the main aim of this paper was to present the design and calibration of a low cost sensors kit (Air Enquirer) and to show the possibility of using it in dynamic flux chambers to reduce the price of possible $CO_2$ flux networks. We noted the lack of a robust metrology chain in these type of measurements and we think it is important to describe and proposed it. However, we also believe that it will be really important to test our $CO_2$ dynamic flux system with other available systems being them commercial or from other research groups. Actually, we are going to submit a new project to, among others goals, buy a new Licor $CO_2$ flux system ([https://www.licor.com/env/products/soil_flux/](https://www.licor.com/env/products/soil_flux/)) and make an intercomparison campaign between different systems. This has been now better clarified in the conclusion of this manuscript as further actions. In addition, we have changed the title of the manuscript to better fit with its content.

Following your comments, we have expanded the calibration section in order to better describe the processes applied for the calibration of the sensors. Moreover, we are now proposing different type of calibrations, we have calculated the errors associated to each one of them and we are doing recommendations for calibration and recalibration of the instruments depending on their specific applications. We think this gives to the study a strong metrology basis.

In the following lines we discussed each one of your comment and we indicate the respective changes, if it is the case, within the manuscript.

*1) L17: You wrote that the results were corrected for illumination. This does not appear anywhere in the main body of the article. Please either remove this or edit section 2.2 and the results section to reflect this. Do you think illumination would have an impact on the CO2 calibration? I could only imagine an indirect effect through heating of the sensor, which is already captured by the temperature calibration.*

Answer: The reviewer is right. Within this study the data of the illumination sensor was not used. This has been clarified in the revised version of the manuscript (Line 17). This sensor was added

at the kit because we thought it would be interesting for phenology and other ecological studies, but it is not used for the calibration of the $CO_2$ sensors.

*2) L88: In my experience employing very similar sensors (SenseAir LP8), air humidity has a large effect on the data quality, particularly for RH> 80, causing a highly nonlinear saturation in the measured concentration. Could you please report the RH range used for calibration? As you measure directly above soil, I would expect the RH in the chamber to routinely reach these values. Did you experience this? This could have an impact on data quality, particularly if you experience a sharp increase in RH within the chamber during the measurement time. Could you show the graph of RH for the linear accumulation experiment of Figure 6?*

Answer: As asked by the reviewer we have added both temperature and relative humidity values measured within the NSS-NTF chamber during the measurement examples shown in Figure 6 (in the revised version is Figure 7). In the NSS-NTF there is always an increase in temperature and relative humidity, but this increase is not sharp, probably thanks to the really short period of time of the measurements. In addition, for the reviewer we added here in Figure A1 a plot with the temperature and RH data from the SSTF experiment of Figure 5 (now Figure 6). We did not observe RH > 80% neither for NSS-NTF or for SSTF chambers.

For the calibration experiments, the RH range was 10%-50%, and the Temperature ranged in the interval 20 $^0$C -42 $^0$C. Plots of this data have been added into the revised version of the manuscript (Figure 4).

[Figure]

**Figure 7. Example of two cases where the linear accumulation method was applied within an NSS-NTF chamber to calculate positive (a) and negative (b) $CO_2$ fluxes with Air Enquirer Kit #03.**

[Figure]

*Figure A1: a) %RH from all sensors in the SSTF chamber b) T from all sensors in the SSTF chamber.*

*3) I appreciate the effort spent to develop a high-quality calibration model. However, in the case of NSS-NTF flux measurements, calibrated data is not strictly necessary as long as the sensor's calibration does not change during the measurement timespan as the flux is determined "differentially" by considering the rate of growth of the CO2 concentration within the chamber. Could you briefly comment on this in your paper? This could save significant resources for researchers that intend to reproduce your system but do not have the means to perform a comprehensive calibration.*

Answer: As commented by the reviewer in the case of NSS-NTF chambers the absolute $CO_2$ concentration value is not used because the need parameter is the slope of the $CO_2$ concentration increase during the time interval. Thus it is only important that the calibration factors of the sensors do not change over the timespan of the measurement. We have added this discussion in the paper as explained in detail in the next point.

*4) In contrast to the above point, in the case of SS-TF chambers, the effect of calibration is not negligible here. For example, if the sensor in the mixing chamber shows a large bias in its response compared to the sensor in the flux chamber (or vice versa), the bias will induce a similar*

*bias in the estimated flux. Therefore, it is clear that uncalibrated sensors cannot be used in this configuration. Could you comment on this? Could you suggest a simplified calibration procedure for those who do not have access to calibration facilities? Would a field calibration with a known gas sample be sufficient?*

Answer: Considering comments 3 and 4, we decided to expand and to improve our study. Now we apply different calibration approaches and compare the RMSE obtained with them: i) raw data; ii) theoretical correction for RH and P; iii) theoretical correction + bias removal; iV) theoretical correction + simple calibration; v) theoretical correction + multiparametric calibration. The whole chapter 3.1 has been rewritten. The RMSE for all kind of calibrations is now showed in table 2. In figure 5, we also show the fit results from the different calibration approximations. Benefiting of these previous results we have added a section in the results and discussion chapter (chapter 3.3) where we make recommendations on how to proceed with the calibration and recalibration of sensors, differentiating between sensors for SS-TF chambers and for NSS-NTF chambers, and purposing different solutions depending on the laboratories possibilities. We have emphasised and quantified the importance of the bias removal in the SS-TF sensors for the flux calculation, and also quantified the error introduced in the NSS-NTF by not calibrating the sensors.

*5) L167 could you provide a summary of the environmental parameters the sensors were exposed to during the calibration cycle? Did you systematically sweep a space of T/RH configurations?*

Answer: In lines 117-121 of the manuscripts the intervals are now detailed: *"Both experiments were performed in a temperature range between 20 ºC and 42 ºC and a relative humidity with diurnal cycles between 10% and 50%. Temperature in the calibration box was set to be in increased in slopes of 10ºC, although at low temperatures it fluctuated with room temperature. The pressure ranged between 1004 hPa and 1012 hPa in the calibration at IC3 and between 838 hPa and 850 hPa in the calibration at CRAM."*. In the new Figure 4 the variations in T and RH during the IC3 and CRAM calibrations are shown.

*6) L175 could you also show the response of the sensors as a timeseries of relative deviation from the reference provided by the CRDS? Showing this timeseries alongside T and RH would provide some insights regarding their effect on data quality as mentioned in question 3.*

Answer: As suggested by the reviewer we have added a plot (Figure 4) with the differences between the CRDS reference and the kit values for simple calibration and multiparametric fitting. In this plot it can be seen that each kit response differently to the variation of T and RH.

*7) L183 you report that pressure has the largest effect of the sensor response and that pressure compensation provides the largest reduction in RMSE. Why did you choose to use a linear model for pressure compensation instead of using the quadratic formula suggested by SenseAir in their application note? (see http://www.co2meters.com/Documentation/AppNotes/AN149-Senseair-Pressure-Dependence.pdf)*

Answer: First of all, we thank the reviewer to indicate us this link. We did not know about this technical sheet. During our study we found out the following documents (https://cdn.shopify.com/s/files/1/0019/5952/files/AN001_Pressure_Compensating_of_a_CO2_Sensor_Rev_1.0_03_May_2021.pdf) where a linear approximation, considering the Ideal Gas Law, was recommended. However, in order to evaluate the reviewer comment, we have made

Pressure tests for all sensors. The plot copied below reports $CO_2$ sensors data in relation to the environmental pressure during the experiments ($\approx$ 1010 hPa and $\approx$ 845 hPa at IC3 and at CRAM sites, respectively). To skip the $CO_2$ concentration influence we filtered the data using only those ones when the $CO_2$ concentration measured by the reference instrument was in the interval 415 ppm - 425 ppm. The plot also shows the quadratic (red line) and lineal (blue line) models calculated using the previous cited references for a concentration of 420 ppm. Bias for $CO_2$ data is corrected for every kit taking as reference the 1010 hPa data. It can be seen that each sensor seems to have a its own Pressure dependency but, generally, data get closer to the linear ideal gas law fit.

On the light of it, what we did was to apply a three steps correction/calibration:

      a.   correction for wet air-dry air;

      b.   correction for Pressure influence following the ideal gas law;

      c.   calibration of the $CO_2$ sensor response using a multiparametric fit.

This process has been better explained within the manuscript now.

[Figure]

*8) L185 While your statement is true in our experience with the CarboSense network; I think such frequent calibrations could be replaced by in-field calibration for intercept/sensitivity using one respectively two references gases. In the case of NSS-NTF measurements, calibration is not strictly necessary to obtain corrected flux measurements as I pointed out above.*

Answer: Authors agree with the reviewer and, actually, a "calibration strategy" paragraph (chapter 3.3) has been now added within the "results and discussion" section where different type of calibrations are recommend in relation to the sensors applications.

*9) L202 For the SS-TF measurements you employed multiple sensors on each chamber. Do you think this redundancy could be exploited to improve data quality, for example by averaging the signals or to determine biases?*

Answer: Authors agree with the reviewer. Actually, the duplicity of the sensors within the chambers was used for improving the reliability of the measurements, in order to cope with malfunctioning sensors but also for averaging and reduce uncertainty. We have added explanation for this in the revised version of the manuscript.

---

## Author Comment (AC2)

Answer reviewer 2:

*This paper presents a new Steady-State-Through-Flow (SS-TF) system based on low-cost Air Enquirer kits, including $CO_2$ and environmental parameter sensors. The $CO_2$ sensor is calibrated in a chamber where environmental parameters can be controlled. Multivariate regression models are derived from comparison with reference $CO_2$ measurements and applied to the $CO_2$ soil flux measurements. Conceptually, this work on application of low-cost sensors for a high temporal and spatial monitoring of $CO_2$ soil flux is useful, but requires more evidence on the performance evaluation of a new SS-TF system to be published in AMT.*

> 2. *Only 5 comparison during 2 days of experiment are provided for the evaluation. This size of dataset is extremely limited. 2 days are not enough to catch all possible range of variations in environmental parameters that might affect the correction of the low-cost $CO_2$ measurements and calculation of the soil flux measurements. Moreover, it would be necessary to have an explanation and a correction for the mismatch observed when NSS-NTF shows negative flux.*
>
> 3. *For the evaluation of this new SS-TF system, I would prefer to see comparison with a commercial soil flux measurement system instead of comparison to NSS-STF measurement system using the low-cost sensor.*

First of all, we want to thank the reviewer for his/her comments. We think his/her experience can help us to improve the quality of the work.

We agree with the reviewer that a full validation of a new SS-TTF system will need a long-term comparison with commercial and/or research products chambers. However, we think it is really important here to underline that the main aim of this paper was not validating a new system but:

> i)   presenting a low cost sensors kit (Air Enquirer);
> ii)  offering a robust metrology for the calibration and the following application of low cost $CO_2$ sensors for environmental studies;
> iii) showing for the first time a methodology of using low cost sensors kit in $CO_2$ fluxes networks, reducing price and improving data quality.

We noted the lack of a robust metrology chain in these type of measurements and we think it is important to describe and propose it.

We also believe that it will be really important to test our $CO_2$ dynamic flux system with other available systems being them commercial as well as from other research groups. Actually, we are going to submit a new project to, among others goals, buying a new Licor $CO_2$ flux system (https://www.licor.com/env/products/soil_flux/) and making an intercomparison campaign between different systems. However, the short comparison exercise carried out and presented in this paper shows that the new SS-TTF systems allows $CO_2$ flux values of the same order of magnitude that the ones observed with a simple static accumulation chamber and in the literature. This is a first important step. All this has been now better clarified in the conclusion of this manuscript as further actions, and in order to avoid misunderstands we have now modified the title of the manuscript for better fitting with its content.

The new version of the manuscript clarifies better the goal of the study and the results presented.

*Specific comments*

*Line 97-100. Detailed description on the calibration chamber system is needed. How is the calibration experiment designed? For example, at what temperatures is the experiment held and for how long?*

Thanks the reviewer for this comment. In order to answer to this comment, we think it is important to clarify what implies the calibration of $CO_2$ low cost sensors and the calibration of $CO_2$ flux chambers. In the first case, as explained within the manuscript, it is need a metrology to calibrate low cost $CO_2$ sensors and to understand the influence of environmental parameters on their response. This has been extensively done within this study and better presented in the revised version of this manuscript. In lines 117-122, the range of temperature, RH and pressure is detailed:*" Both experiments were performed in a temperature range between 20 ℃ and 42 ℃ and a relative humidity with diurnal cycles between 10% and 50%. Temperature in the calibration box was set to be in increased in slopes of 10℃, although at low temperatures it fluctuated with room temperature. The pressure ranged between 1004 hPa and 1012 hPa in the calibration at IC3 and between 838 hPa and 850 hPa in the calibration at CRAM. The two calibration experiments at the CRAM and at IC3 stations were carried out with one month difference."*

We have added a new figure (new Figure 4) where the difference between calibrated sensor and the CRDS $CO_2$ value are plotted together with temperature and humidity values.

In regard to the calibration of flux chambers, this should be done creating a complete metrology chain where a primary reference standard: a $CO_2$ respiration soil is used to calibrate the response of the fluxes systems. An example of it is the metrology chain created by the project traceRadon for radon flux measurement (Roettger et al., 2021).

Fluxes system, as well as other monitors and systems, can be compared between them to carry out proficiency studies and to validate systems results. In this case you do not have any reference but you estimated the participants using a mean value of the participant's response and you can estimate the dispersion between them.

As explained in the introduction of this document the main aim of this study was not validated a new system but design, built a calibrate a new low cost sensors kit and apply it for new application showing it feasibility, low maintenance, low cost and further possible application for the scientific community.

*Line 138. How well would the measurements at the top of the flux chamber represent the gas exiting the chamber? How much bias or uncertainty would be introduced with this assumption?*

Answer: In order to minimize the concentration gradient within each chamber a fan was used to homogenize the air inside. Moreover, two instruments were located in two different point within the chamber to smooth possible bias. Finally, the total uncertainty budget of the $CO_2$ flux measurements has been presented with k=2 to have a bigger coverage factor.

*Line 195-196. Is concentration first averaged and then used to calculate the flux? Or is the flux calculated using the original temporal resolution of the $CO_2$ measurements and then averaged?*

Answer: The flux has been calculated using the original minute $CO_2$ measurement and then its average value has been calculated over 10 minutes. The revised version of the manuscript has been modified (lines 239-243) to clarify this procedure:

*"Each value of flux has been calculated using Eq. (7) and averaging the calibrated $CO_2$ values of AE #1 and #2 for the mixing chamber and taken the data from AE #3 for the flux chamber. 10 min. averages were calculated from every minute calculated flux data. The variability of the flux within the 10 minutes averages is represented in Fig. 6 as an associated uncertainty of $2\sigma$. The associated expanded uncertainty for each value has been calculated propagating the $2*RMSE_{multi}$ of the flux chamber $CO_2$ sensor."*

*Line 197. What is the temporal resolution of the CO2 measurements? Is the RSE also calculated with 10 minutes averaged dataset? If not, the RSE would be different for the 10 minutes averaged flux.*

Answer: The flux is calculated using equations 7 and 8 for dynamic and static chambers respectively. The RMSE is not the uncertainty of the flux but the Root Mean Square Error of the calibration fit of each kit with temporal resolution of 1 minute. The uncertainty of the flux has been calculated propagating the uncertainties of the variable and parameters participating in the Equation 7 and 8, respectively. Then in the case of the dynamic system, the uncertainty of the mean was also propagated for the 10 used values.

The new revised version of the manuscript explains this better now.

*Figure 5. What's the difference between the 2 sigma error and the extended error?*

Answer: The 2-sigma is twice the standard deviation of the flux within the 10 minutes average. The extended error adds the uncertainty associated with the sensors measurement (with k=2) to this variability. Although it was commented in the text, we have clarified (lines 239:243) and we have also added the explanation in the figure caption (now Figure 6):

*"Figure 6. Time series of 10-min average $CO_2$ concentrations (upper panel) measured within the SS-TF chamber at the CRAM soil between 1st and 2nd of June 2016, and calculated $f_{CO_2}$ (lower panel). The $2\sigma$ range for 10 minutes average variability and the extended error (adding 2 times the RSE of the multiparametric fit) are also plot."*